# *CTHRC1* Expression Results in Secretion-Mediated, SOX9-Dependent Suppression of Adipogenesis: Implications for the Regulatory Role of Newly Identified CTHRC1^+^/PDGFR-Alpha^+^ Stromal Cells of Adipose

**DOI:** 10.3390/ijms26051804

**Published:** 2025-02-20

**Authors:** Matthew E. Siviski, Rachel Bercovitch, Kathleen Pyburn, Christian Potts, Shivangi R. Pande, Carlos A. Gartner, William Halteman, Doreen Kacer, Barbara Toomey, Calvin Vary, Robert Koza, Lucy Liaw, Sergey Ryzhov, Volkhard Lindner, Igor Prudovsky

**Affiliations:** 1Center for Molecular Medicine, MaineHealth Institute for Research, Scarborough, ME 04074, USA; matthew.siviski@gmail.com (M.E.S.); rbercovitch2017@gmail.com (R.B.);; 2Graduate School of Biomedical Science and Engineering, University of Maine, Orono, ME 04469, USA; 3Technical and Administrative Services, University of Maine, Orono, ME 04469, USA

**Keywords:** collagen triple helix repeat-containing 1 (CTHRC1), SRY-box transcription factor 9 (SOX9), adipogenesis, adipogenic gene expression, white adipose tissue, stromal vascular cells, PDGFR-alpha

## Abstract

Adipogenesis is regulated by the coordinated activity of adipogenic transcription factors including PPAR-gamma and C/EBP alpha, while dysregulated adipogenesis can predispose adipose tissues to adipocyte hypertrophy and hyperplasia. We have previously reported that *Cthrc1*-null mice have increased adiposity compared to wildtype mice, supporting the notion that CTHRC1 regulates body composition. Herein, we derived conditioned medium from 3T3-L1 cells expressing human *CTHRC1* and investigated its anti-adipogenic activity. This constituent significantly reduced 3T3-L1 cell adipogenic differentiation commensurate to the marked suppression of *Cebpa* and *Pparg* gene expression. It also increased the expression of the anti-adipogenic transcription factor SOX9 and promoted its nuclear translocation. Importantly, *Sox9* gene knockdown demonstrated that the anti-adipogenic effect produced by this conditioned medium is dependent on SOX9 expression, while its ability to positively regulate SOX9 was attenuated by the application of Rho and Rac1 signaling pathway inhibitors. We also identified the selective expression of *CTHRC1* in *PDGFRA*-expressing cell populations in human white adipose tissue, but not brown or perivascular adipose tissues. Congruently, flow cytometry revealed CTHRC1 expression in PDGFR-alpha^+^ stromal cells of mouse white adipose tissue, thus defining a novel stromal cell population that could underpin the ability of CTHRC1 to regulate adiposity.

## 1. Introduction

Whether attributed to genetic predispositions, unhealthy lifestyle choices, or a combination of these risk factors, obesity is the result of an energy imbalance in relation to food (caloric) intake versus energy expenditure, where excess calories are stored as fats within adipocytes [1]. Over time, this energy imbalance causes the production of a greater quantity and size of adipocytes and fat stores, which puts obese individuals at a significantly increased risk of developing obesity-related comorbidities [2]. We previously characterized collagen triple helix repeat-containing 1 (CTHRC1) as a secreted factor whose expression negatively correlates with adiposity [3]. We showed that *Cthrc1*-null mice are characterized by significantly increased adiposity in both subcutaneous and visceral adipose tissues in comparison to age-matched wildtype mice [3]. Moreover, transgenic mice overexpressing human *CTHRC1* had significantly decreased adipocyte size and adiposity in relation to wildtype controls [3]. Collectively, these data provide evidence to support the notion that CTHRC1 regulates adipose tissue hypertrophy and hyperplasia.

We further demonstrated that CTHRC1 suppresses in vitro adipogenesis [3]. Primary stromal vascular cells were isolated from the inguinal white adipose tissue (iWAT) of wildtype or age-matched *Cthrc1*-null mice. Upon the chemical induction of adipogenic differentiation in vitro, stromal cells derived from *Cthrc1*-null mice had comparatively enhanced adipocyte differentiation and maturation [3]. These data were recapitulated using 3T3-L1 preadipocyte cells, in which adenoviral overexpression of human *CTHRC1* significantly inhibited adipogenic differentiation [3]. Independently, Takeshita and colleagues reported that the retroviral overexpression of mouse *Cthrc1* in 3T3-L1 cells also markedly suppressed adipogenesis [4].

Adipogenesis is orchestrated by the coordinated expression of adipogenic transcription factors, including CCAAT/enhancer-binding protein alpha (C/EBP alpha) and peroxisome proliferator-activated receptor gamma (PPAR-gamma), which positively regulate preadipocyte-to-adipocyte differentiation [5]. The C/EBP proteins display discrete, temporal activity during adipogenic differentiation. C/EBP beta and C/EBP delta are expressed early in the preadipocyte-to-adipocyte differentiation program and serve to transactivate C/EBP alpha and PPAR-gamma gene expression [6]. C/EBP alpha and PPAR-gamma then positively regulate the expression of lipogenic genes, including fatty acid binding protein 4 (FABP4), which function to equip the developing adipocyte with the requisite machinery to store, regulate, and metabolize lipids [7]. Herein, we overexpressed human *CTHRC1* in 3T3-L1 cells followed by collection of the resultant conditioned medium (which we refer to as hCTHRC1 conditioned medium). In this report, we demonstrate that hCTHRC1 conditioned medium negatively regulates the mRNA and protein expression levels of adipogenic transcription factors corresponding to the inhibition of preadipocyte-to-adipocyte differentiation. We also report that the expression of SRY-box transcription factor 9 (SOX9) is indispensable to the anti-adipogenic function of hCTHRC1 conditioned medium, while Rho and Rac1 signaling is shown to be required for hCTHRC1 conditioned medium to enhance SOX9 protein expression and SOX9 nuclear translocation. Lastly, we demonstrate that CTHRC1 is expressed in PDGFR-alpha^+^ stromal cells within human and mouse subcutaneous white adipose tissues, which may indicate a role for this newly identified stromal cell population in the regulation of adiposity and body composition [8].

## 2. Results

**hCTHRC1 Conditioned Medium Suppresses Lipid Accumulation in Differentiating Adipocytes**. We have previously reported that adenoviral overexpression of human CTHRC1 significantly inhibits chemically induced 3T3-L1 mouse cell adipogenic differentiation [3]. Herein, we assessed whether CTHRC1 might possess paracrine or juxtacrine anti-adipogenic function by conducting an adipogenesis co-culture experiment. Briefly, 3T3-L1 cells were transduced with adenoviral vectors overexpressing either human *CTHRC1* or control *β*–*galactosidase* (*βgal*). In parallel, non-transduced 3T3-L1 cells were fluorescently labeled with CellTracker. Two co-cultures were then seeded: (1) non-transduced 3T3-L1 cells plus transduced 3T3-L1 cells overexpressing βgal; and (2) non-transduced 3T3-L1 cells plus transduced 3T3-L1 cells overexpressing human CTHRC1 (hCTHRC1). After establishing the co-cultures, adipogenic differentiation was chemically induced. Relative to the βgal co-culture, the percentage of non-transduced cells positively stained with the neutral lipid dye, BODIPY, was significantly lower in the CTHRC1 co-culture after four days of chemically induced adipogenic differentiation (Figure 1A–C).

Next, we prepared conditioned media from 3T3-L1 cells transduced with adenoviral vectors expressing either human *CTHRC1* or control *βgal*, which we refer to as hCTHRC1 or βgal conditioned medium, respectively. Despite divergence in the amino acid composition of the N-terminal signal peptide and the adjacent pro-peptide regions, mouse and human CTHRC1 share nearly identical sequence homology, differing in a single amino acid within the C-terminal domain [9]. In 3T3-L1 cells, endogenous mouse CTHRC1 was not detected at the mRNA or protein levels. The presence of recombinant human CTHRC1 in the conditioned medium was confirmed by an established ELISA or by Western blot analysis (Figure 1D). For these adipogenesis experiments, 3T3-L1 cells were chemically stimulated to undergo adipogenic differentiation for a total period of six days in the continued presence of βgal or hCTHRC1 conditioned medium (refer to Application of Conditioned Medium and Adipogenic Differentiation Timeline), and then stained with the neutral lipid dye, Oil Red O (Figure 1E,F). Based on absorbance spectroscopy measuring the relative concentration of eluted Oil Red O, we observed that 3T3-L1 cells treated with hCTHRC1 conditioned medium had significantly lower lipid content in comparison to cells treated with βgal conditioned medium control (Figure 1E,F).

**hCTHRC1 Conditioned Medium Downregulates Adipogenic Gene Expression**. Given that the application of hCTHRC1 conditioned medium decreased lipid accumulation in 3T3-L1 cells following the induction of adipogenic differentiation (Figure 1E), we next assessed the effect of hCTHRC1 conditioned medium on adipogenic gene expression. As before, 3T3-L1 cells were seeded on Day-3 in the presence of either βgal or hCTHRC1 conditioned medium and then chemically stimulated to undergo adipogenesis beginning on Day 0. Whole-cell lysates were collected on Days 0, 2, and 6 for qPCR and Western blot analyses. The samples collected on Day 0 were undifferentiated controls. Consistent with literature detailing the temporal expression patterns of both adipogenic and lipogenic genes in the course of adipogenic differentiation [6], elevated *Cebpb* and *Cebpd* gene expression on Day 2 preceded significant induction of *Cebpa*, *Pparg*, and *Fabp4* gene expression on Day 6 (Figure 2A–E). These temporal adipogenic gene expression trends were recapitulated at the protein level (Figure 3A–F), in which C/EBP beta and C/EBP delta protein expression were highest on Day 2, while C/EBP alpha, PPAR-gamma, and FABP4 each displayed maximal protein expression levels on Day 6. Cells treated with hCTHRC1 conditioned medium displayed significantly decreased *Cebpd* gene expression on Day 0 (Figure 2B), as well as a marked decrease in *Cebpa*, *Pparg*, and *Fabp4* gene expression on Day 6 (Figure 2C–E). C/EBP alpha, PPAR-gamma, and FABP4 protein expression levels were also significantly reduced on Day 6 in cells treated with hCTHRC1 conditioned medium (Figure 3D–F). Taken together, the ability of hCTHRC1 conditioned medium to suppress lipid accumulation is commensurate to its inhibition of adipogenic gene expression.

**The Anti-Adipogenic Effect of hCTHRC1 Conditioned Medium Is Dependent on SOX9 Expression**. Given evidence in support of the anti-adipogenic function of hCTHRC1 conditioned medium (Figure 1 and Figure 2), we next focused on potential CTHRC1 effector proteins with demonstrated anti-adipogenic activity. It has been shown that a subpopulation of activated fibroblasts in heart tissue co-express high levels of CTHRC1 and SOX9 following myocardial infarction [10]. SOX9 is a well-established transcription factor with marked anti-adipogenic function [11]. When we assessed whether hCTHRC1 conditioned medium regulates *Sox9* gene expression in 3T3-L1 cells, we found that cells treated with hCTHRC1 conditioned medium expressed increased SOX9 at both the mRNA (Figure 4A) and protein (Figure 4B,C) levels in comparison to cells treated with βgal conditioned medium. In addition, confocal microscopy analysis of SOX9 protein localization in 3T3-L1 cells revealed that application of hCTHRC1 conditioned medium significantly enhanced SOX9 nuclear translocation on Day 0 in comparison to cells treated with βgal conditioned medium (Figure 4D,E). Consistent with the anti-adipogenic role of SOX9, which is a transcriptional regulator known to translocate to the nucleus and suppress the activity of adipogenic gene promoters [12], enhanced SOX9 nuclear localization was also observed on Day 4 of chemically stimulated adipogenic differentiation in 3T3-L1 cells treated with hCTHRC1 conditioned medium but not in cells treated with βgal conditioned medium (Appendix A).

We then questioned whether SOX9 is required for the anti-adipogenic activity of hCTHRC1 conditioned medium and used an RNA interference (RNAi) strategy to knock down *Sox9* gene expression in 3T3-L1 cells. Briefly, 3T3-L1 cells were transduced with lentiviral vectors overexpressing either an shRNA construct against mouse *Sox9* mRNA (i.e., shSOX9), or a non-targeting, scrambled control shRNA construct (i.e., shSCR). Puromycin was then used to select transduced shSOX9 and shSCR cells. It is well documented that when primary adipocyte progenitor cells or transformed cell lines, including 3T3-L1 cells, are extensively passaged or become confluent during passaging, the cells lose their ability to efficiently differentiate into adipocytes in vitro [13,14]. Accordingly, we did not implement single-cell cloning strategies following lentiviral transduction and instead propagated all puromycin-selected shSCR and shSOX9 cells in a sub-confluent manner prior to study. Given that hCTHRC1 conditioned medium significantly enhanced *Sox9* mRNA expression (Figure 4A), we further rationalized that higher concentrations of hCTHRC1 conditioned medium could “outcompete” the constitutively expressed shRNA construct targeting *Sox9* mRNA, particularly in shSOX9 cells with less efficient lentiviral transduction, and thus negate overall *Sox9* gene knockdown. On this basis, we first determined an approximate dilution range where hCTHRC1 conditioned medium displayed significant anti-adipogenic activity based on Oil Red O absorbance spectroscopy data (Appendix A), following which we opted to assay βgal and hCTHRC1 conditioned media at a dilution of 1/60 for the RNAi adipogenic differentiation studies. Based on Oil Red O absorbance spectroscopy data, and relative to βgal conditioned medium control, application of hCTHRC1 conditioned medium at a 1/60 dilution significantly decreased 3T3-L1 adipogenic differentiation in shSCR cells but not shSOX9 cells (Figure 4F). Based on the quantification of SOX9 protein expression levels, in comparison to the application of βgal conditioned medium, hCTHRC1 conditioned medium at a 1/60 dilution significantly enhanced SOX9 levels in shSCR cells but not in shSOX9 cells, the latter displaying significant knockdown of SOX9 protein expression levels (Figure 4G,H).

Taken together, our data demonstrate that hCTHRC1 conditioned medium regulates multiple facets of SOX9 signaling: in addition to its ability to significantly enhance *Sox9* gene expression and SOX9 protein expression levels, hCTHRC1 conditioned medium also positively regulates SOX9 nuclear translocation. Importantly, our RNAi adipogenic differentiation studies support the hypothesis that SOX9 protein expression is critical for the anti-adipogenic activity of hCTHRC1 conditioned medium.

**Rho-Like GTPase Signaling Is Required for the Anti-Adipogenic Effect of hCTHRC1 Conditioned Medium**. The anti-adipogenic function of SOX9 is predicated on its nuclear translocation and DNA binding abilities [11]. Specifically, SOX9 has been shown to bind to the promoter regions of adipogenic genes, thus inhibiting their expression and the overall advancement of preadipocyte-to-adipocyte differentiation [12]. We therefore questioned what facets of CTHRC1 signaling might regulate SOX9 nuclear translocation. Recently, Hironaka and colleagues reported that the actin binding protein drebrin stabilizes the F-actin cytoskeleton in myofibroblasts which, in turn, enhances the nuclear translocation of SOX9 [15]. The disruption of actin stress fibers in adipocyte progenitor cells has been shown to enhance their adipogenic differentiation [16], and we have observed that CTHRC1 overexpression enhances the F-actin cytoskeleton in 3T3-L1 cells (Appendix A). Relatedly, Rho-like GTPases are also well characterized regulators of the actin cytoskeleton, wherein several investigations have demonstrated that CTHRC1 overexpression significantly increases the levels of both Rho-GTP and Rac1-GTP [17,18]. Thus, we hypothesized that Rho and Rac1 signaling positively regulate the ability of hCTHRC1 conditioned medium to enhance SOX9 nuclear translocation.

To address this hypothesis, 3T3-L1 cells were seeded in the presence of βgal or hCTHRC1 conditioned medium, with or without the combined treatment of NSC 23766 (“N”) and Y-27632 (“Y”). NSC 23766 is a well-defined Rac1 activation-specific inhibitor [19], while Y-27632 is a potent inhibitor of the direct Rho effector, Rho-associated kinase (ROCK) [20]. Significantly, the combined application of N (10 µM, final) and Y (15 µM, final) attenuated SOX9 nuclear translocation caused by treatment with hCTHRC1 conditioned medium (Figure 5A–E). We next investigated whether the combined application of N and Y also negates the anti-adipogenic effect produced by hCTHRC1 conditioned medium. Similar to the RNAi studies (Figure 4), we applied βgal and hCTHRC1 conditioned media at dilutions of 1/60, in which 3T3-L1 cells were treated with or without N and Y. However, the combined application of N and Y at final concentrations of 10 µM and 15 µM, respectively, resulted in a significant detachment of cells several days after the onset of chemically stimulated adipogenic differentiation. Therefore, we reduced the final concentrations of N and Y to 3 µM and 5 µM, respectively, and repeated the adipogenic differentiation experiments. Based on Oil Red O absorbance spectroscopy data, in three independent experiments, hCTHRC1 conditioned medium produced a significant anti-adipogenic effect when applied in the presence of N and Y (Figure 5F). However, when quantifying the percent suppression of Oil Red O staining relative to the application of hCTHRC1 conditioned medium, the application of N and Y significantly reduced the anti-adipogenic effect of hCTHRC1 conditioned medium (Figure 5G). This result was further corroborated based on two-way analysis of variance (Appendix A), which indicated that hCTHRC1 conditioned medium produced a statistically greater anti-adipogenic effect when N and Y were omitted. In correlating these Oil Red O data relative to SOX9 protein expression levels, it is noteworthy that, in direct comparison to cells treated with βgal conditioned medium, the application of hCTHRC1 conditioned medium with or without the combined presence of N and Y (3 µM and 5 µM, respectively) significantly increased SOX9 protein expression levels (Figure 5H,I). Critically, however, in assessing the effect of hCTHRC1 conditioned medium on SOX9 expression, the application of N and Y significantly reduced SOX9 protein expression levels in comparison to vehicle control (Figure 5I).

The data presented here support that the anti-adipogenic effect of hCTHRC1 conditioned medium is dependent on its positive regulation of Rho and Rac1 signaling. These collective results suggest that both Rho and Rac1 signaling are integral components with respect to the ability of hCTHRC1 conditioned medium to positively regulate SOX9 protein expression, as well as SOX9 nuclear translocation, thus culminating in the suppression of preadipocyte-to-adipocyte differentiation. Quite intriguingly, however, when recombinant human CTHRC1 was removed from hCTHRC1 conditioned medium using immunoprecipitation strategies, the resultant conditioned medium was still able to inhibit 3T3-L1 cell adipogenic differentiation based on Oil Red O staining (Appendix A) and increase SOX9 protein expression levels by Western blot analysis (Appendix A). With respect to the methodology by which we generated hCTHRC1 conditioned medium, these data could support that the expression of human *CTHRC1* in 3T3-L1 cells positively regulates the secretion of an anti-adipogenic factor(s). Regarding a putative CTHRC1-Rho/Rac1-SOX9 axis of signaling, these collective data further support the notion that there exists a CTHRC1 effector present in hCTHRC1 conditioned medium that inhibits adipogenesis by enhancing the protein expression levels and anti-adipogenic activity of SOX9.
Figure 5**Rho and Rac1 Signaling Mediate the SOX9-Dependent, Anti-Adipogenic Function of hCTHRC1 Conditioned Medium In Vitro**. (**A**–**E**) 3T3-L1 cells were seeded on Day-3 with βgal or hCTHRC1 conditioned medium at a 1/4 dilution in the absence or presence of the combined application of NSC 23766 (N; 10 µM) and Y-27632 (Y; 15 µM) (N+Y). (**A**–**D**) Representative confocal microscopy images of SOX9 protein localization on Day 0 in 3T3-L1 cells treated with either βgal conditioned medium (**A**,**C**) or hCTHRC1 conditioned medium (**B**,**D**) in the absence (**A**,**B**) or presence (**C**,**D**) of the combined application of N (10 µM) and Y (15 µM): nuclei (blue); SOX9 (green); F-actin/Alexa Fluor 546 Phalloidin (red). Scale bar: 20 μm. (**E**) Quantification of SOX9^+^ nuclei based on 10 separate fields per experiment (n = 3; ** *p* ≤ 0.01). Not statistically significant (ns). (**F**–**I**) 3T3-L1 cells were seeded on Day-3 with βgal or hCTHRC1 conditioned medium at a dilution of 1/60, in the absence or presence of the combined application of N (3 µM) and Y (5 µM) (N+Y). Whole-cell lysates were collected from cohorts of cells on Day 0 to assess SOX9 protein expression levels by Western blot analysis (**H**), while the other cohorts were chemically stimulated to undergo adipogenic differentiation for a total period of 6 days (**F**). (**F**) Representative Oil Red O quantification data. 3T3-L1 cells were formalin fixed on Day 6 and stained with Oil Red O, which was then eluted and its concentration determined by absorbance spectroscopy. Per experiment, 3T3-L1 cells were plated in 24-well plates in which 6 wells each were treated with βgal or hCTHRC1 conditioned medium at a dilution of 1/60 in the presence or absence of N+Y (n = 4; ** *p* ≤ 0.01, *** *p* ≤ 0.001, **** *p* ≤ 0.0001). (**G**) Graphical representation quantifying that hCTHRC1 conditioned medium has a significantly diminished anti-adipogenic effect when applied in the presence of N+Y (displayed as average percent reduction in Oil Red O staining relative to the application of hCTHRC1 conditioned medium). Data reflect the averages from four independent experiments (n = 4; ** *p* ≤ 0.01). Blue symbols (triangle, square, circle, and diamond) are paired according to experimental replication. (**H**) Representative Western blot image where the vertical line denotes the splice junction within both SOX9 and GTF2B immunoblots. (**I**) Average SOX9 protein fold change densitometry values relative to housekeeping GTF2B protein expression levels from four independent experiments (n = 4; ** *p* ≤ 0.01, *** *p* ≤ 0.001). Green symbols (triangle, square, open circle, and closed circle) are paired according to experimental replication.
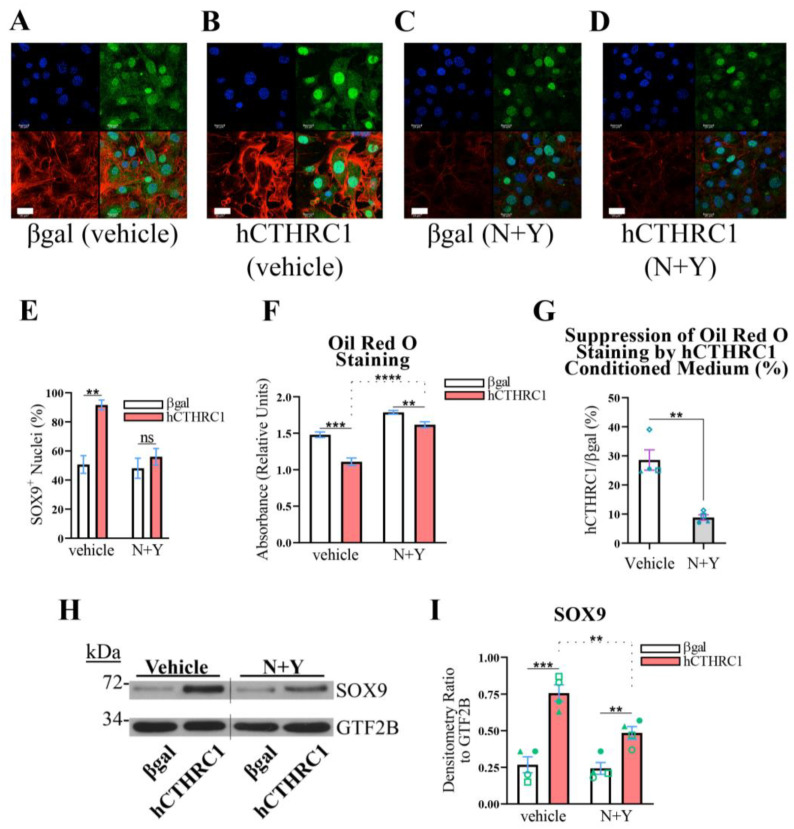


**Identification of PDGFR-alpha^+^:CTHRC1^+^ Stromal Cells in White Adipose Tissue**. We have previously shown that *Cthrc1*-null mice are characterized by significantly increased adiposity in both subcutaneous and visceral adipose tissues in comparison to age-matched wildtype mice [3], supporting the concept that CTHRC1 could restrict adipocyte hyperplasia and hypertrophy and thus regulate in vivo adipogenesis. Moreover, we have also reported that stromal vascular fraction (SVF) cells harvested from mouse inguinal white adipose tissue (iWAT) express detectable *Cthrc1* mRNA levels [3]. Given that the SVF of white adipose tissue contains adipocyte progenitor cells capable of undergoing adipogenesis in vivo [11], we sought to determine the stromal cell population that expresses CTHRC1. A potentially important caveat when assessing CTHRC1 expression in white adipose tissue is the finding from transcriptomic analysis of C57BL/6 mice that *Cthrc1* gene expression in iWAT decreases during early postnatal development (Figure 6A; [21]). Congruently, we observed a marked decrease in *Cthrc1* gene expression in the iWAT isolated from 3-month-old mice versus 7-day-old mice (Figure 6B). On this basis, we proceeded to analyze *Sox9* gene expression in the iWAT isolated from 7-day-old wildtype and *Cthrc1*-null C57BL/6 littermate mice, and observed significantly higher *Sox9* gene expression levels among the iWAT derived from wildtype pups (Figure 6C). Next, we conducted multi-parameter flow cytometry (Figure 6D–J) to assess CTHRC1 protein expression among SVF cells harvested from the iWAT of juvenile C57BL/6 mice (5-day-old pups). Following the strategy of Gulyaeva and colleagues [11], SVF cells that expressed either TER119 (erythroid cells), CD31 (endothelial cells), or CD45 (immune cells) were excluded from the multi-parameter analysis that assessed CTHRC1 expression chiefly among CD24^+^ or PDGFR-alpha^+^ SVF cells. Accordingly, CTHRC1 was detected in PDGFR-alpha^+^ SVF cells (Figure 6I) with negligible CTHRC1 expression detected in CD24^+^ SVF cells (Figure 6J).

We also evaluated *CTHRC1* gene expression in adult human subcutaneous white adipose tissues, brown adipose tissue (BAT), and perivascular adipose tissue (PVAT). Based on available single-nuclei or single-cell RNA sequencing databases, we observed significant *CTHRC1* expression in human subcutaneous white adipose tissues, principally among *PDGFRA*^+^:*MFAP5*^+^ gene-rich cell populations (Figure 7A–D). While *CTHRC1* expression in subcutaneous white adipose tissues did not differ in lean versus obese human subjects, it was not significantly detectable in human BAT or PVAT (Figure 7C). Furthermore, *SOX9* mirrored the gene expression patterns of *CTHRC1*, albeit expressed at lower levels, and was most abundantly expressed in human subcutaneous white adipose tissues among *PDGFRA*^+^:*MFAP5*^+^ gene-rich cell populations (Figure 7D). Taken together, these data support that CTHRC1 is expressed among PDGFR-alpha^+^ stromal cells in human and mouse subcutaneous white adipose tissues (Figure 8), in which our analysis of the iWAT depot in mice further underscores the age-dependent, temporal pattern of *Cthrc1* gene expression therein.
Figure 6**PDGFR-alpha^+^ Stromal Cells Express CTHRC1 In Vivo**. (**A**) *Cthrc1*, *Mfap5*, and *Pdgfra* mRNA expression levels in inguinal white adipose tissue (iWAT) during postnatal development. Koza and colleagues [21] conducted microarray analyses of RNA isolated from the iWAT of C57BL/6 male mice aged 5, 10, 21, and 56 days, in which *Cthrc1* and *Mfap5* gene expression were shown to decrease in iWAT during early postnatal development. (**B**) qPCR analysis of *Cthrc1* gene expression in iWAT isolated from 7-day-old and 3-month-old C57BL/6 male mice. Data were normalized to housekeeping *Gtf2b* expression levels and are presented as the average fold value per mouse (n = 4; ** *p* ≤ 0.01). Black diamonds refer to the relative *Cthrc1* fold value per mouse. (**C**) qPCR analysis of *Sox9* gene expression in iWAT isolated from 7-day-old wildtype and *Cthrc1*-null C57BL/6 littermate mice that were derived from heterozygous breeding pairs (three litters in total). Data were normalized to housekeeping *Gtf2b* expression levels and are presented as the average fold value per mouse (n = 5; * *p* ≤ 0.05). Black symbols (triangle, circle, diamond, open square, and closed square) are paired according to experimental replication comparing a littermate wildtype mouse to a *Cthrc1*-null mouse. (**D**–**J**) Representative multi-parameter flow cytometry workflow, displayed in the form of contour plots, assessing endogenous CTHRC1 protein expression in CD24^+^ versus PDGFR-alpha^+^ stromal vascular fraction (SVF) cells. SVF cells were harvested from the iWAT of 5-day-old wildtype and *Cthrc1*-null C57BL/6 mice. SVF cells from three mice were pooled together per genotype (n = 4). Dead SVF cells, in addition to SVF cells of erythroid lineage (TER119^+^), CD31^+^ endothelial cells, and CD45^+^ immune cells (i.e., lineage-negative cells), were omitted from the multi-parameter analysis. (**D**,**E**) VioBlue-treated SVF cells stained without (**D**) or with (**E**) lineage-negative markers. Side scatter (SCC) area. (**F**,**G**) Live SVF cells treated without (**F**) or with (**G**) antibodies against CD24 and PDGFR-alpha. (**H**) No detectable CTHRC1 expression in PDGFR-alpha^+^ SVF cells derived from *Cthrc1*-null mice. (**I**) Detectable CTHRC1 expression in certain PDGFR-alpha^+^ SVF cells derived from age-matched wildtype mice. (**J**) Negligible expression of CTHRC1 in CD24^+^ SVF cells derived from age-matched wildtype mice.
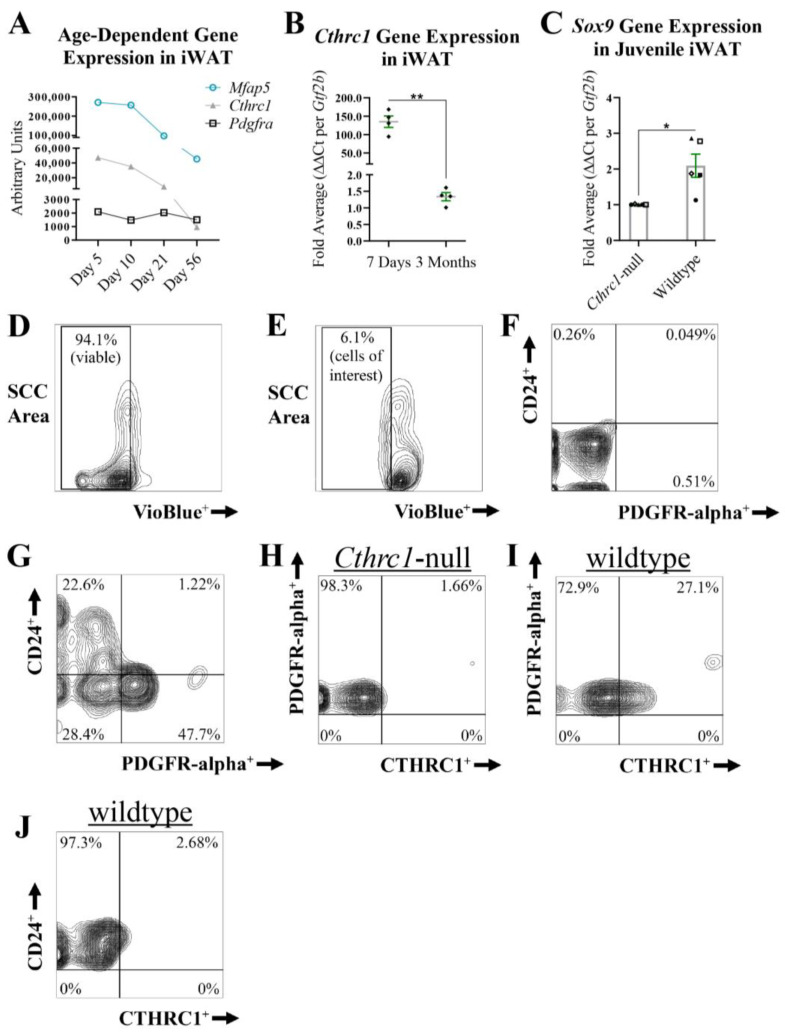


## 3. Discussion

In this report, we investigated the anti-adipogenic activity of conditioned medium derived from 3T3-L1 cells expressing human *CTHRC1*. As such, we demonstrated that hCTHRC1 conditioned medium negatively regulates adipogenic transcription factor gene expression at both the mRNA and protein levels. hCTHRC1 conditioned medium also positively regulated *Sox9* gene expression and the nuclear localization of SOX9 protein. Congruent with the anti-adipogenic function of SOX9, wherein SOX9 silences adipogenic genes by binding to their promoter regions in an inhibitory manner [11,12], hCTHRC1 conditioned medium preserved SOX9 nuclear localization even after the chemical stimulation of adipogenic differentiation in vitro. This spatiotemporal observation supports that SOX9 nuclear translocation is a critical element with respect to the anti-adipogenic effect of hCTHRC1 conditioned medium. We also identified the contribution of Rho-like GTPase signaling relative to the ability of hCTHRC1 conditioned medium to suppress adipogenesis. Specifically, we observed that the combined suppression of Rho and Rac1 signaling attenuates the ability of hCTHRC1 conditioned medium to enhance SOX9 protein expression as well as SOX9 nuclear localization, thus resulting in a significant reduction in the anti-adipogenic activity of hCTHRC1 conditioned medium.

We further defined that SOX9 protein expression is indispensable to the anti-adipogenic effect of hCTHRC1 conditioned medium based on adipogenic differentiation studies that implemented RNAi-mediated *Sox9* gene knockdown in 3T3-L1 cells. The 3T3-L1 cell line, which constitutes the most extensively investigated in vitro model of adipogenesis [24], has been shown to display a diminished capacity to differentiate into adipocytes if extensively passaged or reaches confluence during passaging [13,14]. For this reason, as part of our RNAi-based studies, we did not implement single-cell cloning strategies following the lentiviral transduction and puromycin selection of shSCR and shSOX9 cells. Despite extensively optimizing our lentiviral transduction methods, including the applied concentration of lentiviral particles for each vector, in addition to refining the duration of transduction preceding puromycin selection, we consistently observed that more shSCR cells survived puromycin selection than shSOX9 cells. Thus, at the moment shSCR and shSOX9 cells were seeded for study, shSOX9 cells were at a slightly higher passage than shSCR cells. This is reflected in our Oil Red O absorbance spectroscopy data revealing that shSCR cells underwent an overall higher degree of adipogenic differentiation than shSOX9 cells. However, from these RNAi adipogenic differentiation studies, it was consistently observed that the anti-adipogenic effect of hCTHRC1 conditioned medium is dependent on SOX9 protein expression levels. In addition, the chemical suppression of Rho and Rac1 signaling pathways not only enhanced the adipogenic differentiation of 3T3-L1 cells despite concomitant treatment with hCTHRC1 conditioned medium, but also significantly attenuated the ability of hCTHRC1 conditioned medium to enhance SOX9 protein expression. These results are congruent with Gulyaeva and colleagues who demonstrated that *Sox9*-deficient, primary mouse adipocyte progenitor cells underwent increased adipogenic differentiation in vitro in comparison to *Sox9*-wildtype adipocyte progenitor cells [11], thus supporting the overall correlation between enhanced adipogenesis and decreased SOX9 expression.

Collectively, our in vitro results support that a putative CTHRC1-Rho/Rac1-SOX9 signaling axis negatively regulates adipogenesis. The finding that the immunodepletion of recombinant human CTHRC1 from hCTHRC1 conditioned medium does not affect the anti-adipogenic capacity of the conditioned medium itself is intriguing. These data could support the notion that there exists a secreted CTHRC1 effector present in hCTHRC1 conditioned medium that inhibits adipogenesis by enhancing anti-adipogenic SOX9 signaling. Future investigations focused on the molecular characterization of hCTHRC1 conditioned medium are necessary to identify such a putative CTHRC1 effector. Given the robust anti-adipogenic effect of hCTHRC1 conditioned medium, it is conceivable that there exist multiple CTHRC1 effectors that positively regulate discrete aspects of SOX9 gene expression and SOX9 protein nuclear translocation. Such CTHRC1 effectors could harbor increased expression levels within hCTHRC1 conditioned medium or perhaps even exhibit specific post-translational modifications that augment anti-adipogenic SOX9 signaling. Indeed, future investigations are required to define how CTHRC1 regulates the secretome relative to its ability to enhance SOX9 signaling and suppress adipogenesis.

The results of multi-parameter flow cytometry analysis identified a novel PDGFR-alpha^+^:CTHRC1^+^ stromal cell population retained within inguinal white adipose tissue (iWAT). During early postnatal development, adipose tissue compartments are significantly expanding due to several factors: adipocyte progenitor cells are actively recruited to differentiate into adipocytes, and there is a simultaneous increase in adipocyte hypertrophy [25]. From lineage tracing investigations of stromal vascular fraction (SVF) cells in iWAT, mesenchymal stems cells define the earliest adipocyte progenitor cell population and are characterized by PREF1^+^:CD24^+^:PDGFR-alpha^-^ expression [11]. Ultimately, mesenchymal stem cells advancing in the adipogenic lineage become PREF1^−^:CD24^−^:PDGFR-alpha^+^ preadipocytes capable of undergoing terminal differentiation to white adipocytes [11]. Our finding that CTHRC1 is expressed among CD24^−^:PDGFR-alpha^+^ stromal cells in mouse iWAT could therefore suggest that CTHRC1 is expressed in the adipogenic lineage among a population of preadipocytes.

In a recent study focused on the gene expression profiles of human adipose tissue-derived SVF cells, *CTHRC1* mRNA levels were significantly enhanced following the chemical induction of adipogenic differentiation of these SVF cells in vitro [26]. However, *CTHRC1* mRNA expression was upregulated in an extracellular matrix gene-rich subpopulation that did not terminally differentiate into mature adipocytes [26]. Relatedly, to identify cells expressing *CTHRC1* in human adipose tissues, we queried single-cell and single-nuclei transcriptomic data, and report herein the selective expression of *CTHRC1* in *PDGFRA*^+^:*MFAP5*^+^ gene-rich cell populations in subcutaneous white adipose tissues, but not within brown adipose tissue or perivascular adipose tissue. In the context of adipogenesis, Vaittinen and colleagues found that microfibrillar-associated protein 5 (MFAP5) localizes in the extracellular matrix of human abdominal subcutaneous adipose tissue [27]. Using an in vitro model of human adipogenic differentiation, it was reported that *MFAP5* mRNA is expressed maximally in preadipocytes and minimally in adipocytes [27]. These data are congruent with a recent investigation demonstrating that MFAP5 suppresses 3T3-L1 cell adipogenic differentiation [28]. Although MFAP5 has been characterized as a secreted protein associated with microfibrils within the extracellular matrix [27], Zhang and colleagues reported an intracellular functionality in its suppression of adipogenesis given that MFAP5 was shown to both bind to and inhibit the expression of SND1 (staphylococcal nuclease and Tudor domain-containing 1), which is a well-established co-activator of PPAR-gamma [28]. Intriguingly, similar to the gene expression profile of *Cthrc1* in mouse iWAT, *Mfap5* mRNA levels also decrease during early postnatal development, suggesting that the downregulation of both *Cthrc1* and *Mfap5* gene expression could be required for terminal differentiation of preadipocytes in vivo. Moreover, it is also interesting to note that among human subcutaneous white adipose tissues, relative *CTHRC1* gene expression levels in *PDGFRA*^+^:*MFAP5*^+^ gene-rich stromal cell populations do not significantly vary in comparing lean versus obese adult subjects. As with mouse models, it will be important to determine *CTHRC1* gene expression trends in white adipose tissue during human development and whether diminished *CTHRC1* gene expression in early postnatal development is predictive of juvenile, adolescent, or adult obesity. In an obesogenic context, it would also be important to determine whether specific *CTHRC1* gene mutations correlate with human obesity, particularly those mutations that could potentially hamper the anti-adipogenic activity of CTHRC1 or its effectors.

MFAP5 expression has also been reported in various fibroblast populations including cancer-associated fibroblasts [29]. Similarly, although PDGFR-alpha is expressed in preadipocytes [11], it is a known marker of stromal fibroblast populations that are separate from the adipogenic lineage [30]. Regarding stromal cells resident in mouse white adipose tissues, Marcelin and colleagues found that PDGFR-alpha^+^:CD9^high^ stromal populations possess a myofibroblast-like phenotype in which *Cthrc1* mRNA expression significantly correlated with this PDGFR-alpha^+^ sub-population robustly expressing CD9 protein [31]. Correspondingly, *Mfap5* gene expression was enhanced in this PDGFR-alpha^+^:CD9^high^ stromal cell population [31]. *Cthrc1* mRNA expression has also been detected in mouse cardiac fibroblasts [10,15]. In one report, both *Cthrc1* and *Sox9* gene expression were significantly upregulated in a subpopulation of activated, pro-fibrotic fibroblasts following myocardial infarction in mice [10]. More recently, Hironaka and colleagues demonstrated using mouse cardiac myofibroblasts that the actin binding protein drebrin stabilizes the F-actin cytoskeleton, increases *Cthrc1* gene expression, and promotes the nuclear translocation of SOX9 protein [15]. Therefore, this corroborating investigation suggests that F-actin stability, in relation to SOX9 localization within the nucleus, comprises core elements of CTHRC1-related signaling.

Among white adipose tissues, our data present the possibility that CTHRC1 is expressed in the adipogenic lineage in a population of preadipocytes and could function to suppress preadipocyte-to-adipocyte differentiation in vivo. On the other hand, CTHRC1 may also be expressed in adipose PDGFR-alpha^+^ stromal fibroblasts, from where it could regulate the adipogenic differentiation of resident adipocyte progenitor cells. Accordingly, refined lineage tracing experiments will be essential to determine the adipose tissue expression of CTHRC1 among discrete fibroblast and/or adipocyte progenitor cell populations, and to further define the role of CTHRC1 in the regulation of in vivo adipogenesis in states of metabolic health versus disease.

Adipose tissue hypertrophy and hyperplasia are well defined characteristics of obesity, in which obese individuals are at a greater risk of developing diabetes, metabolic syndrome, and other related comorbidities [32]. In the obese state, there are multiple lines of evidence supporting that adipocytes can become insulin resistant [33]. In the lean state, adipocyte turnover, in which old adipocytes are degraded and new adipocytes arise owing to adipogenic differentiation, has also been demonstrated to be an essential facet of maintaining insulin-responsive adipocytes [34]. Future investigations will be critical to elucidate the molecular mechanism by which CTHRC1 regulates adipogenic signaling. Thus, this mechanistic knowledge could form the basis for therapeutically targeting CTHRC1 to finely regulate the rate at which new, insulin-responsive adipocytes are formed, while altogether preventing precocious adipogenesis in vivo.

## 4. Materials and Methods

*Adenoviral Transduction and Conditioned Media Preparation*: 3T3-L1 cells, originally from Zen-Bio (Durham, NC, USA), were seeded at passage 14 in DMEM (high glucose, without sodium pyruvate; Sigma Aldrich, St. Louis, MO, USA; D5796) supplemented with 4% fetal bovine serum (*v*/*v*; Hyclone, Logan, UT, USA; SH30396.03), 6% bovine calf serum (*v*/*v*; Cytiva, Marlborough, MA, USA; SH30072.03), and 1× antibiotic-antimycotic solution (Gibco, Waltham, MA, USA; 15240-062), which we refer to as full-serum DMEM, in a 5% CO_2_ atmosphere at 37 °C. Cells were transduced as described [3] with adenoviral vectors overexpressing either human *CTHRC1* or control *β*–*galactosidase* (*βgal*) for 8 h, and then incubated for 15 h in 22 mL of full-serum DMEM. Conditioned media were centrifuged at 450× *g* for 3 min to pellet any detached cells, after which the supernatants were collected. The presence of recombinant human CTHRC1 in conditioned medium was confirmed by Western blot analysis or by an established ELISA as described previously [3]. From ELISA, we determined that the concentration of recombinant human CTHRC1 in the collected conditioned medium was approximately 9–10 ng/mL. In this study, we evaluated the effect of human CTHRC1 conditioned medium at the indicated dilutions prepared in full-serum DMEM. 

*Exogenous Application of Conditioned Media and Adipogenic Differentiation*: 3T3-L1 cells were grown and expanded in a sub-confluent manner in 10 cm dishes in full-serum DMEM. For our study, 3T3-L1 cells were seeded at passage 11 at a density corresponding to 70% confluence in the presence of freshly diluted βgal or human CTHRC1 conditioned medium and were vigorously agitated at regular intervals for 2 h to prevent cell aggregation in the center of the wells. We designated this initial subconfluent seeding of 3T3-L1 cells as Day-3. Following this, media were changed on a daily basis with freshly diluted βgal or human CTHRC1 conditioned medium. Three days after the initial seeding, on a day we designated as Day 0, cells were treated with adipogenic differentiation induction cocktail including insulin (10 μg/mL, final; Sigma Aldrich, St. Louis, MO, USA; 19278), 3-isobutyl-1-methylxanthine (IBMX, 0.5 mM, final; Sigma Aldrich, St. Louis, MO, USA; 410957), and dexamethasone (1 μM, final; Tocris, Bristol, UK; 1126), and then replenished with identical, freshly prepared media on Day 1. On Days 2 and 4, media were changed and replenished with insulin (10 μg/mL, final). On Days 3 and 5, media were changed and replenished with only βgal or human CTHRC1 conditioned medium. In studies utilizing the Rac1 inhibitor, NSC 23766 (Tocris, Bristol, UK; 2161), and the Rho-associated kinase inhibitor, Y-27632 (Tocris, Bristol, UK; 1254), these reagents were freshly added daily beginning on Day-3. A diagrammatic representation of our methodology is presented immediately below.



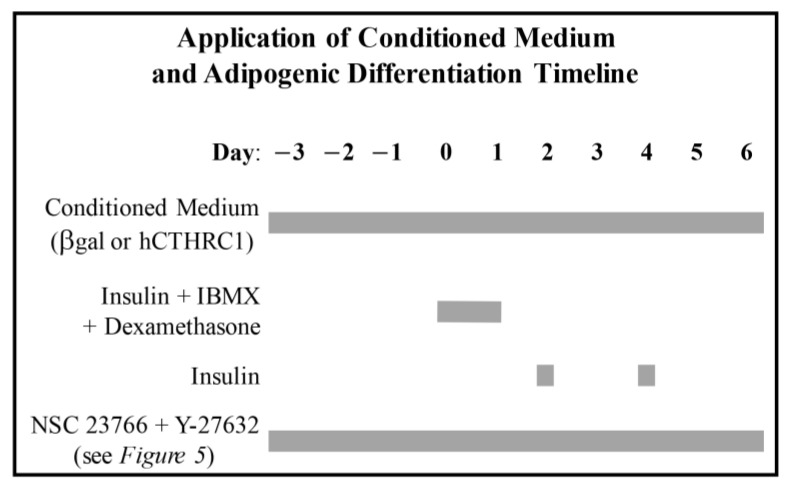



*Assessment of SOX9 Nuclear Localization*: 3T3-L1 cells were seeded in triplicate on sterile 12 mm glass coverslips in the presence of βgal or human CTHRC1 conditioned medium, as described above. At the indicated timepoints, cells were fixed using 10% neutral formalin (Sigma Aldrich, St. Louis, MO, USA; F5554) and blocked overnight at 4 °C in immunofluorescence blocking buffer consisting of PBS supplemented with 2% bovine serum albumin (*w*/*v*), 0.2% Tween 20 (*v*/*v*), 0.1% sodium azide (*w*/*v*), and 0.2% Triton X-100 (*v*/*v*). Following blocking, primary antibodies against SOX9 (Cell Signaling Technology, Danvers, MA, USA; Cat. # 82630) were added at a 1/250 dilution in immunofluorescence blocking buffer and kept overnight at 4 °C. The next day, Alexa Fluor 546 Phalloidin (Molecular Probes, Eugene, OR, USA; A22283) and Alexa Fluor 488 chicken anti-rabbit antibody (Invitrogen, Waltham, MA, USA; A21441) were each prepared at a 1/250 dilution along with Hoechst 33342 (4 µM, final; Thermo Fisher Scientific, Waltham, MA, USA; 62249), added to the cells, and incubated in darkness for 1 h at room temperature. The cells were then embedded in Vectashield Mounting Medium (Vector Laboratories, Inc., Newark, CA, USA; H-1000) and analyzed by confocal microscopy using the Leica SP8 microscope (Leica Microsystems, Wetzlar, Germany) within the Histopathology and Microscopy Core Facility at MaineHealth Institute for Research.

*Lentiviral-Mediated RNA Interference Studies*: Lentiviral plasmids expressing a shRNA construct targeting *Sox9* mRNA (i.e., shSOX9: CTCCACCTTCACTTACATGAA) or a non-targeting scrambled control shRNA construct (i.e., shSCR: CCTAAGGTTAAGTCGCC-CTCG) were a kind gift from Dr. Robert Weinberg [35] and cloned in the pLKO.1-puro lentiviral plasmid vector at the Viral Vector Core Facility at MaineHealth Institute for Research. For lentiviral transduction, passage 9 3T3-L1 cells were seeded in two 10 cm dishes at approximately 600,000 cells per dish in full-serum DMEM. 4 h later, cells were transduced with shSOX9 or shSCR lentivirus for 15 h with 8 µg/mL of polybrene (Sigma Aldrich, St. Louis, MO, USA; TR-1003-G). Following lentiviral transduction, cells were continually selected and expanded in full-serum DMEM supplemented with puromycin (2 µg/mL, final; Gibco, Waltham, MA, USA; A11138-03). For our studies, shSOX9 and shSCR cells were seeded in the presence of βgal or human CTHRC1 conditioned medium as described above.

*Oil Red O Staining*: Cells were fixed with 10% neutral formalin overnight at room temperature and then washed with 60% 2-propanol. Oil Red O (Sigma Aldrich, St. Louis, MO, USA; 1320-06-5) solution was prepared and added to the cells as previously described [3] and incubated on a rotator at room temperature for 15 min. Oil Red O was eluted from the stained cells with 100% 2-propanol, and the absorbance was then measured at 520 nm using a FlexStation 3 plate reader (Molecular Devices, San Jose, CA, USA). Oil Red O absorbance data were normalized by subtracting the average absorbance value of 2-propanol from the raw absorbance data.

*Western Blotting*: Whole-cell lysates were prepared from cells by scraping each TC24 well in 50 µL of protein lysis buffer (ITSI Biosciences, Johnstown, PA, USA; K-0045-50). Lysates from respective samples were pooled together. Relative protein concentrations were determined by absorbance spectroscopy using Coomassie Brilliant Blue (Thermo Scientific, Waltham, MA, USA; 23236) measured at 595 nm. Lysates were then resuspended with 4× Laemmli Sample Buffer supplemented with 10% 2-mercaptoethanol, boiled at 100 °C for 5 min, and loaded at a volume of 20 µL per lane in 12% polyacrylamide gels at equalized protein concentrations alongside a broad range protein standard (New England BioLabs, Ipswich, MA, USA; P7719S). Blots were blocked in Tris-buffered saline with Tween 20 (TBST; Cell Signaling Technology, Danvers, MA, USA; 9997) supplemented with 5% powdered, non-fat milk (*w*/*v*) for 2 h at room temperature or overnight at 4 °C, and then incubated with primary antibody overnight in TBST supplemented with 2% bovine serum albumin and 0.1% sodium azide. The following day, secondary antibodies were prepared to a final dilution of 1/5000 in TBST supplemented with 5% powdered, non-fat milk (*w*/*v*), after which the blots were incubated for 1 h at room temperature. Chemiluminescent solution (ProSignal Pico; Prometheus, San Diego, CA, USA; 20-300B) was applied to the blot for development by autoradiography. The following rabbit primary antibodies were purchased from Cell Signaling Technology (Danvers, MA, USA): C/EBP beta (Cat. # 3087), C/EBP delta (Cat. # 2318), C/EBP alpha (Cat. # 8178), PPAR-gamma (Cat. # 2435), FABP4 (Cat. # 2120), SOX9 (Cat. # 82630), and GTF2B (Cat. # 4149), which served as the housekeeping protein control for each experimental replication. Rabbit anti-BSA (bovine serum albumin) antibody was purchased from Rockland Immunochemicals (Limerick, PA, USA; Cat. # 101-4133). Mouse monoclonal antibody (clone Vli19C07) raised against the N-terminus of human CTHRC1 was produced in-house: further product information is available on the MaineHealth Institute for Research website. The following HRP-conjugated secondary antibodies were purchased from Cell Signaling Technology (Danvers, MA, USA): horse anti-mouse (Cat. # 7076) and goat anti-rabbit (Cat. # 7074).

*RT-qPCR*: RNA was prepared from cell culture by scraping each TC24 well in 50 µL of TRIzol Reagent (Ambion, Waltham, MA, USA; 15596018). Lysates from respective samples were pooled together. Subsequently, the volume of TRIzol was readjusted to 1 mL per sample. For studies involving the isolation of RNA from mouse inguinal white adipose tissue, frozen adipose tissue was pulverized in liquid nitrogen and then resuspended in 1 mL of TRIzol. Next, 100 µL of stock 1-bromo-3-chloropropane (Sigma Aldrich, St. Louis, MO, USA; B9673) was added to a 1 mL volume of TRIzol/lysed cells per sample. Each tube was allowed to sit for 5 min at room temperature, followed by a 15 min centrifugation at 12,000× *g* at 4 °C. 350 µL of the upper aqueous layer was then mixed with 350 µL of 70% ethanol and transferred to a Zymo-Spin IIICG Column included in the *Quick*-RNA Miniprep Kit (Zymo Research, Irvine, CA, USA; R1055). RNA was subsequently isolated following the kit manufacturer’s instructions. The RNA concentration per sample was measured using a NanoDrop 2000c spectrophotometer (Thermo Fisher Scientific, Waltham, MA, USA; ND2000CLAPTOP), and 500 ng of RNA was converted to cDNA following the manufacturer’s instructions (Reverse Transcription Supermix for RT-qPCR; Bio-Rad, Hercules, CA, USA; 1708841). cDNA was loaded at 5.7 ng per well in a 96-well qPCR plate (USA Scientific, Ocala, FL, USA; 1402-8900) in triplicate per sample in a 20 µL final reaction volume. Respective forward and reverse cDNA primer pairs (see below) were loaded in each well at a final concentration of 839 nM plus AzuraQuant Green (Azura Genomics, Raynham, MA, USA; 2024-07) according to the manufacturer’s instructions. The following 5′-to-3′ forward (F) and reverse (R) primer sequences were used to detect the following target mouse genes:

*Sox9* (F: CACACGTCAAGCGACCCATGAA;

R: TCTTCTCGCTCTCGTTCAGCAG),

*Gtf2b* (F: ATGGCGGACAGAATCAACCTCC; 

R: ACAAGCAGAGGCTATCGCGTCA), 

*Cthrc1* (F: CCTGGACCCCAAACTATAAGCA; 

R: AGCCACTGAACAGAACTCGC),

*Cebpb* (F: CAACCTGGAGACGCAGCACAAG; 

R: GCTTGAACAAGTTCCGCAGGGT), 

*Cebpd* (F: CGAGAACGAGAAGCTGCATCAG; 

R: CCCAAAGAAACTAGCGATTCGG), 

*Cebpa* (F: GCAAAGCCAAGAAGTCGGTGGA; 

R: CCTTCTGTTGCGTCTCCACGTT), 

*Pparg* (F: GTACTGTCGGTTTCAGAAGTGCC; 

R: ATCTCCGCCAACAGCTTCTCCT), 

*Fabp4* (F: GCTGCAGCCTTTCTCACC;

R: CACTTTCCTTGTGGCAAAGC).

*Co-Culture*: Passage 11 3T3-L1 cells were transduced with adenoviral vectors overexpressing either human *CTHRC1* or control *β*–*galactosidase* as described above. In parallel, non-transduced 3T3-L1 cells at passage 11 were grown in full-serum DMEM. Upon reaching approximately 70% confluence, non-transduced cells were incubated for 30 min in a 2 µM solution of CellTracker Deep Red (Thermo Fisher Scientific, Waltham, MA, USA; C34565) and then trypsinized. The CellTracker-labeled, non-transduced 3T3-L1 cells were then added to trypsinized human CTHRC1- or βgal-transduced cells in equal numbers and plated on 22 × 22 mm sterile glass coverslips at 750,000 cells per well in a TC6 plate in full-serum DMEM. The media were replenished after 24 h. Cells were then chemically stimulated to undergo adipogenesis beginning on Day 0 following our standard protocol as described above. On Day 4, cells were fixed using 10% neutral formalin and blocked overnight at 4 °C using immunofluorescence blocking buffer (see above). Cells were incubated with Hoechst 33342 (4 µM, final), Bodipy 493/503 (5 µM, final; Invitrogen, Waltham, MA, USA; D3922), and Alexa Fluor 546 Phalloidin (1/100 final dilution) prepared in immunofluorescence blocking buffer (see above) for 1 h at room temperature in darkness, embedded in Vectashield Mounting Medium, and analyzed by confocal microscopy.

*Assessment of the F-actin Cytoskeleton*: Passage 11 3T3-L1 cells were seeded on sterilized 22 × 22 mm glass coverslips in triplicate and were then transduced with adenoviral vectors overexpressing either human *CTHRC1* or control *β*–*galactosidase* as described above. Two days after transduction, cells were fixed using 10% neutral formalin and blocked overnight at 4 °C in immunofluorescence blocking buffer (see above). Following blocking, Hoechst 33342 (4 µM, final) and Alexa Fluor 546 Phalloidin (1/100 final dilution) were prepared in immunofluorescence blocking buffer and added to the cells for 1 h at room temperature in darkness, after which each coverslip was embedded in Vectashield Mounting Medium and analyzed by confocal microscopy.

*Immunodepletion of Human CTHRC1 From hCTHRC1 Conditioned Medium*: In a 15 mL conical tube, 0.45 g of Protein A Sepharose powder (Cytiva, Marlborough, MA, USA; 17078001) was fully resuspended in 2 mL of distilled water by vortexing. Next, an additional 10 mL volume of distilled water was added. The slurry was mixed by vortexing and centrifuged for 3 min at 450× *g* following which the supernatant was aspirated. This washing procedure using a 10 mL volume of distilled water was repeated three additional times, after which the slurry was resuspended in PBS to a final volume of 6 mL and stored at 4 °C. 1 mL of this PBS/Protein A Sepharose slurry mixture was then transferred to two 1.5 mL microcentrifuge tubes each and incubated for 1 h by rotation at 4 °C with either 16 µL of mouse anti-CTHRC1 IgG (produced in-house, clone Vli13E09) at a stock concentration of 0.6 mg/mL or naïve mouse IgG (Sigma Aldrich, St. Louis, MO, USA; 12-371) at an equivalent concentration. Respective Protein A Sepharose-IgG conjugates were then transferred to separate 15 mL conical tubes each containing 10 mL of PBS, mixed by vortexing, and then centrifuged for 3 min at 450× *g* following which the supernatant was aspirated. This washing procedure using a 10 mL volume of PBS was repeated once more, after which each respective Protein A Sepharose-IgG conjugate was resuspended in 12 mL of PBS and transferred to two new 15 mL conical tubes at a volume of 6 mL per tube and then centrifuged for 3 min at 450× *g*, following which the supernatant was aspirated. Next, respective Protein A Sepharose-IgG conjugates were resuspended in 13 mL of either hCTHRC1 or βgal conditioned medium, gently mixed by inversion, and placed upright at 4 °C. The tubes were gently mixed by inversion every 10 min for a total duration of 30 min and then centrifuged for 3 min at 450× *g*. Respective conditioned media supernatants were then transferred to new 15 mL conical tubes and incubated with freshly prepared Protein A Sepharose-IgG conjugates. The entire immunodepletion procedure was repeated three times in total, following which the conditioned media supernatants were stored at 4 °C. An established ELISA confirmed a greater than 80% removal of human CTHRC1 protein levels from hCTHRC1 conditioned medium treated with Protein A Sepharose/mouse anti-CTHRC1 IgG conjugate. Of note, prior to incubation with respective Protein A Sepharose-IgG conjugates, hCTHRC1 and βgal conditioned medium were first diluted three times using serum-free DMEM to enhance the efficiency of human CTHRC1 immunoprecipitation from hCTHRC1 conditioned medium by reducing the relative concentration of sera. For study, conditioned media were assayed at a 1/60 final dilution for which the final percentages (*v*/*v*) of fetal bovine serum and bovine calf serum were readjusted to their standard 4% and 6%, respectively.

*Mice*: *Cthrc1*-null mice with global, homozygous inactivation of the collagen triple helix repeat-containing 1 gene have been previously described [36]. Wildtype and *Cthrc1*-null mice on the C57BL/6 background were kindly provided by Dr. Volkhard Lindner.

*Isolation of Stromal Vascular Fraction Cells*: Inguinal white adipose tissue was surgically removed (both left and right lobes) per mouse, minced with a sterile razor blade, and digested with collagenase D (1.5 units/mL; Roche, Basel, Switzerland, CH; 59983422) and dispase II (2.4 units/mL; Sigma Aldrich, St. Louis, MO, USA; D4693) in 2.5 mL of DMEM/F12 (Corning, Corning, NY, USA; 10-092-CV) containing 0.8% bovine serum albumin at 37 °C for 45 min with agitation. Dissociated cells were passed through a 100 μm strainer and centrifuged at 450× *g* for 3 min. The cells were then resuspended in 1 mL of PBS solution supplemented with 0.5% bovine serum albumin and 2 mM EDTA (ethylenediaminetetraacetic acid; Sigma Aldrich, St. Louis, MO, USA; E9884) which we refer to as FACS buffer.

*Multi-Parameter Flow Cytometry*: The isolated stromal vascular fraction cells (see above) were treated with 1 mL of ACK lysis buffer (ThermoFisher Scientific, Waltham, MA, USA; A1049201) for 5 min at room temperature in 15 mL conical tubes. Next, 10 mL of FACS buffer was added and the cells were centrifuged at 450× *g* for 3 min. Cells were resuspended in 200 µL of VioBlue viability stain (ThermoFisher Scientific, Waltham, MA, USA; L3495), which was prepared in DPBS (Corning, Corning, NY, USA; 21-031-CV) according to the manufacturer’s instructions, and transferred to a FACS tube (Falcon, Corning, NY, USA; 352054). Cells were light protected and incubated for 25 min at room temperature, washed in 1 mL of DPBS, and centrifuged for 3 min at 450× *g*. Cells were resuspended in 100 µL of Fc blocking solution (TruStain FcX, BioLegend, San Diego, CA, USA; 101320) according to the manufacturer’s instructions for 10 min at 4 °C. 250 µL of FACS buffer was then added, and cells were distributed to three FACS tubes at a volume of 100 µL/tube. Next, 100 µL of FACS buffer was added to each tube supplemented with or without the following antibody panels—panel 1: no antibodies; panels 2 and 3: CD24^FITC^ (BioLegend, San Diego, CA, USA; 101816; clone M1/69), PDGFR-alpha^PeCy7^ (BioLegend, San Diego, CA, USA; 135912; clone APA5), CD31^PacificBlue^ (BioLegend, San Diego, CA, USA; 102422; clone 390), CD45^PacificBlue^ (BioLegend, San Diego, CA, USA; 103126; clone 30-F11), and TER119^PacificBlue^ (BioLegend, San Diego, CA, USA; 116232). Cells were incubated for 30 min at 4 °C, after which FACS buffer was added at 1.5 mL/tube, followed by centrifugation for 3 min at 450× *g* and supernatant aspiration. Cells were incubated in fixation/permeabilization buffer (BD Biosciences, Woburn, MA, USA; 554714) for 40 min at room temperature, followed by washing with permeabilization solution (BD Biosciences, Woburn, MA, USA; 554714) at 1 mL/tube, centrifugation for 4 min at 450× *g*, and supernatant aspiration. Next, 200 µL of permeabilization solution was added to each tube supplemented with or without the following antibody panels—panels 1 and 2: no antibodies; panel 3: CTHRC1^APC^ (produced in-house, clone Vli08G09). Cells were incubated for 30 min at 4 °C, after which permeabilization solution was added at 2 mL/tube, followed by centrifugation for 4 min at 450× *g* and supernatant aspiration. Cells were resuspended in FACS buffer at 250 µL/tube. Panels were sequentially analyzed using the MACSQUANT Analyzer (Miltenyi Biotec, Bergisch Gladbach, Germany) following the manufacturer’s instructions at the Flow Cytometry Core Facility at MaineHealth Institute for Research.

*RNA Sequencing*: Human PVAT isolation, library preparation, sequencing, and analytical methodologies were conducted by Angueira and colleagues [37]. Filtered feature barcode matrices were retrieved from the Gene Expression Omnibus. Previously published [30] deep neck BAT single-nucleus RNA-seq filtered feature barcode matrices were retrieved from the European Bioinformatics Institute and reanalyzed. Single-cell RNA sequencing was conducted on human subcutaneous adipose tissue [23] and filtered feature barcode matrices were retrieved from the Gene Expression Omnibus. See Data Availability Statement for accession numbers.

Seurat objects were constructed for each tissue sample (n = 18) using Seurat v4.1.1 [38]. Data were filtered based on number of unique features, percent.mt, and doublets were removed using Scrublet v1.0. Filtered objects were integrated together using the library harmony v0.1.1. Data were log normalized with a scale factor of 10,000 using the Seurat function NormalizeData. For use in clustering, an assay was added in which the normalized data were scaled to fit a distribution with a variance of 1 and a mean of 0 using the ScaleData function. The variables percent.mt, nFeature_RNA, and the S and G2M cell cycle scores, determined through the CellCycleScoring, were regressed out to limit effects on clustering. A principal component analysis was performed using the Seurat function RunPCA over features which were identified as highly variable through use of the previously ran FindVariableFeatures function. Dimensionality reduction was accomplished using the FindNeighbors, FindClusters, and RunUMAP functions with the dims and resolution parameters set to 60 and 0.6, respectively. The DimPlot, FeaturePlot, and VlnPlot functions were utilized to visualize the clusters and marker genes. Manual cluster identification was executed using the FindAllMarkers function, while automatic cluster identification was supplementally performed using the ScType v1.0 package. Immune cells were removed based on PTPRC and MRC1 expression. To save the plots, ggsave function from ggplot2 v3.3.6 was utilized [39].

*Statistical Analysis*: A paired Student t-test was used to compare the mean values of two conditions. In addition, two-way analysis of variance was used to compare the “vehicle” group to the “N+Y” group presented in Figure 5F. All data in this report are presented as the mean ± SEM (standard error of the mean) and were calculated using GraphPad Software (version 10.0.1). A *p*-value of <0.05 was considered statistically significant for all our experimental analyses.

## Figures and Tables

**Figure 1 ijms-26-01804-f001:**
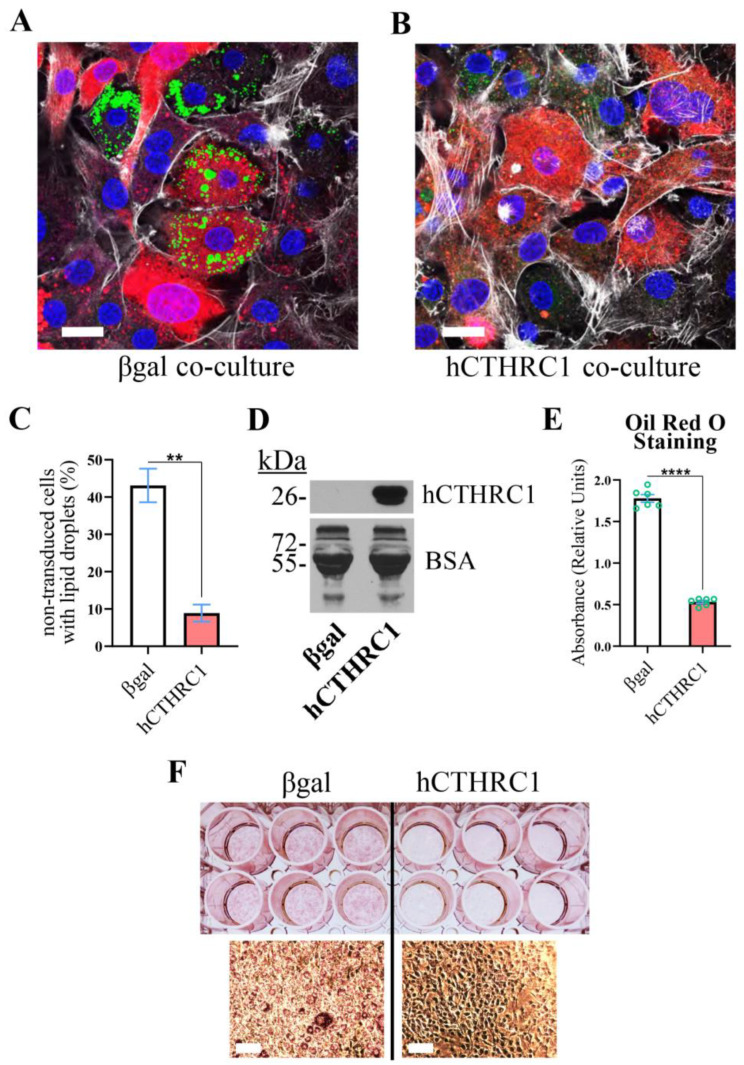
**hCTHRC1 Conditioned Medium Suppresses Lipid Accumulation In Vitro**. (**A**,**B**) Representative confocal microscopy images of non-transduced 3T3-L1 cells labeled with CellTracker Deep Red (red) that were co-cultured with unlabeled 3T3-L1 cells adenovirally transduced with control *β*–*galactosidase* (**A**) or human C*THRC1* (**B**). Each co-culture group was subjected to chemically induced adipogenic differentiation for four days, and then stained with the neutral lipid dye, Bodipy (green). Nuclei were stained with Hoechst (blue), and the F-actin cytoskeleton was stained with Alexa Fluor 546 Phalloidin (white). Scale bar: 10 μm. (**C**) Quantification of CellTracker-labeled cells stained with Bodipy per co-culture. Ten separate fields were analyzed per co-culture per experiment (n = 3; ** *p* ≤ 0.01). (**D**) Western blot analysis of hCTHRC1 protein expressed in hCTHRC1 conditioned medium but not βgal conditioned medium. Bovine serum albumin (BSA) was used as a loading control. The predicted molecular weight of BSA is 66 kDa. (**E**,**F**) Representative Oil Red O quantification data. 3T3-L1 cells were seeded on Day-3 with βgal or hCTHRC1 conditioned medium at a 1/4 dilution, and then chemically stimulated to undergo adipogenic differentiation for a total period of 6 days. Cells were formalin fixed on Day 6 and stained with Oil Red O, which was then eluted and its concentration determined by absorbance spectroscopy (**E**) (n = 3; **** *p* ≤ 0.0001). Per experiment, 3T3-L1 cells were plated in one 24-well plate in which 6 wells were treated with either βgal or hCTHRC1 conditioned medium. Green symbols (circles) represent technical replicates from one experiment. (**F**) Representative image of Oil Red O staining: (*left*) cells treated with βgal conditioned medium; (*right*) cells treated with hCTHRC1 conditioned medium; (*lower panel*) representative Oil Red O staining at 15× magnification. Scale bar: 100 μm.

**Figure 2 ijms-26-01804-f002:**
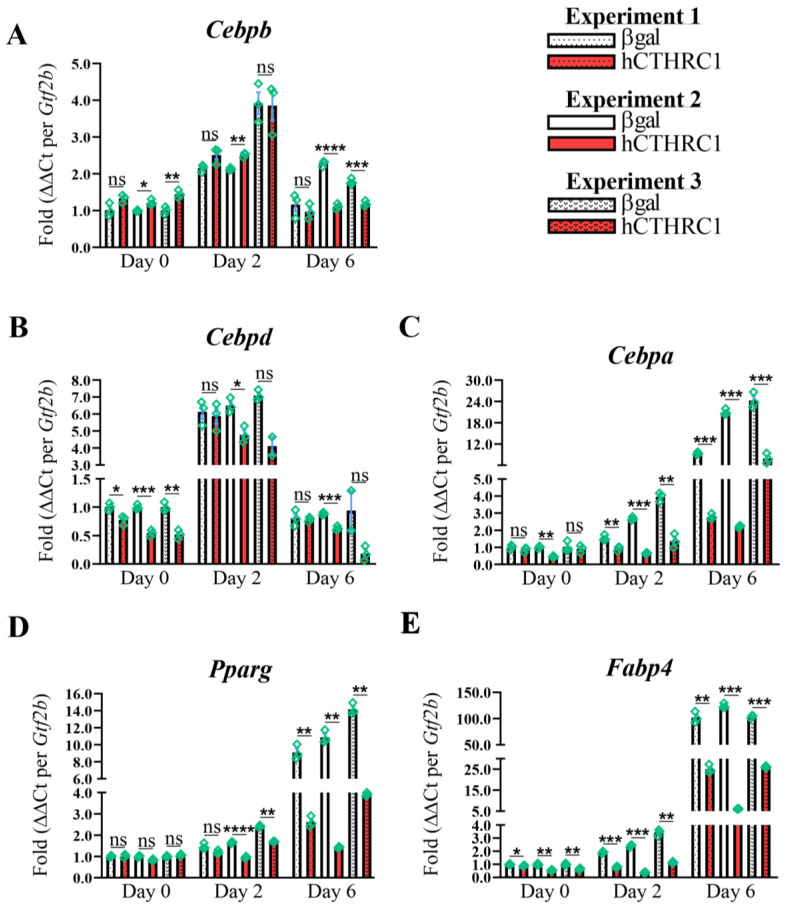
**Exogenously Applied hCTHRC1 Conditioned Medium Inhibits Adipogenic Gene Expression In Vitro**. (**A**–**E**) hCTHRC1 conditioned medium applied to 3T3-L1 cells before and during the course of chemically stimulated adipogenic differentiation negatively regulates the mRNA expression of adipogenic and lipogenic factors as demonstrated by qPCR analyses. 3T3-L1 cells were seeded on Day-3 with either βgal or hCTHRC1 conditioned medium at a 1/4 dilution, after which whole-cell lysates were collected on either Day 0, 2, or 6 relative to the chemical induction of adipogenic differentiation. (**A**–**E**) qPCR fold expression differences in specific mRNA transcript levels relative to housekeeping *Gtf2b* expression levels from three independent experiments (n = 3; * *p* ≤ 0.05, ** *p* ≤ 0.01, *** *p* ≤ 0.001, **** *p* ≤ 0.0001). Green symbols (diamonds) represent technical replicates per experiment. Not statistically significant (ns).

**Figure 3 ijms-26-01804-f003:**
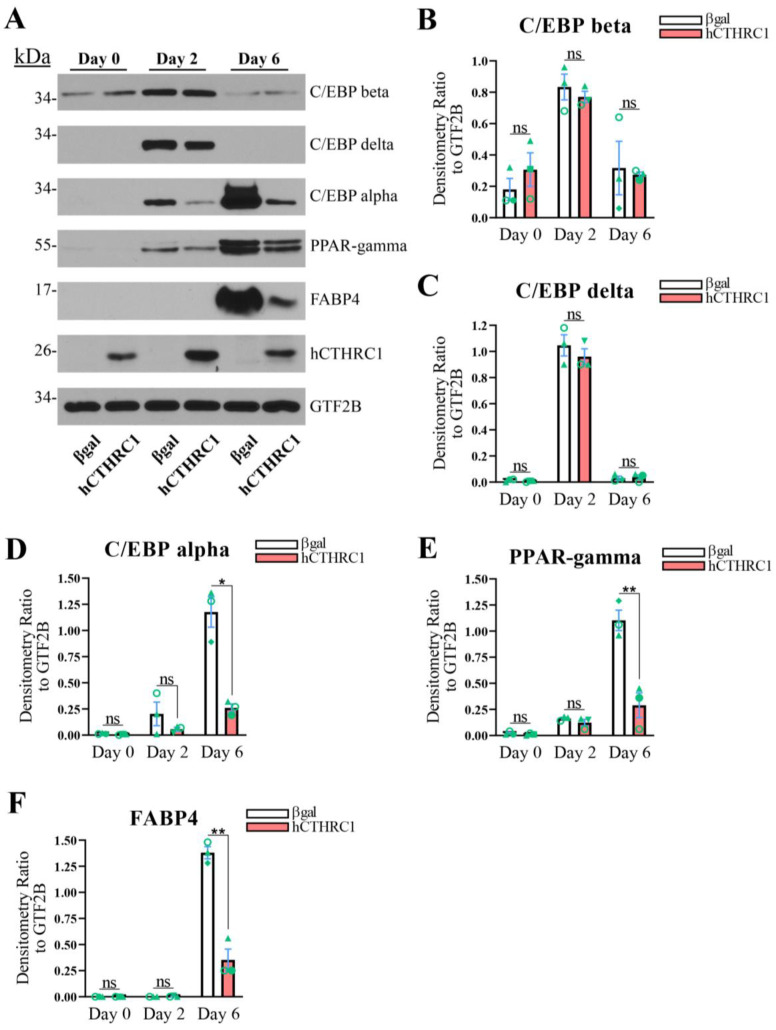
**Exogenously Applied hCTHRC1 Conditioned Medium Inhibits Adipogenic Transcription Factor Protein Expression In Vitro**. (**A**–**F**) hCTHRC1 conditioned medium applied to 3T3-L1 cells before and during the course of chemically stimulated adipogenic differentiation negatively regulates the protein expression of adipogenic and lipogenic factors as demonstrated by Western blot analyses. 3T3-L1 cells were seeded on Day-3 with either βgal or hCTHRC1 conditioned medium at a 1/4 dilution, after which whole-cell lysates were collected on either Day 0, 2, or 6 relative to the chemical induction of adipogenic differentiation. (**A**) Representative Western blots. (**B**–**F**) Average protein fold change densitometry values relative to housekeeping GTF2B protein expression levels from three independent experiments (n = 3; * *p* ≤ 0.05, ** *p* ≤ 0.01). Green symbols (triangle, circle, and diamond) are paired according to experimental replication. Not statistically significant (ns).

**Figure 4 ijms-26-01804-f004:**
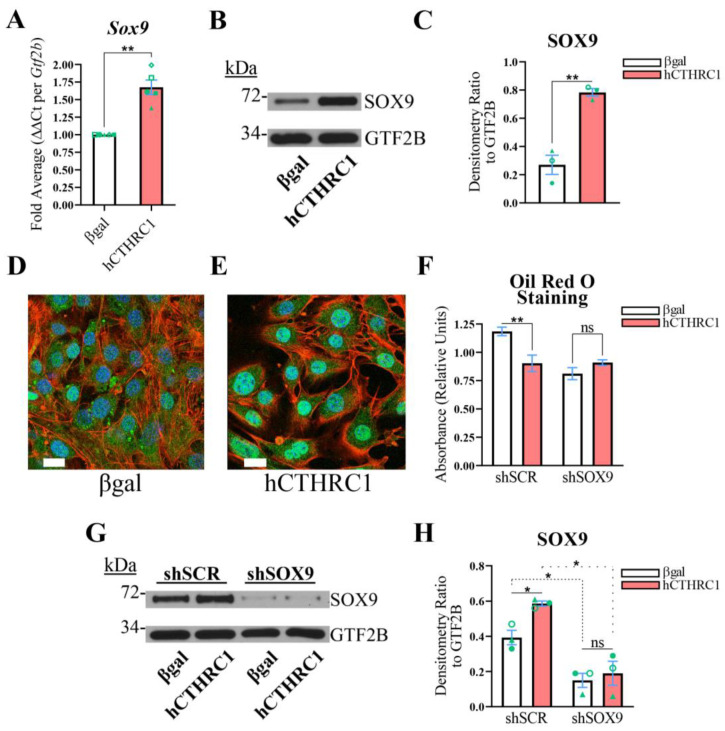
**SOX9 Expression Is Indispensable to the Anti-Adipogenic Activity of hCTHRC1 Conditioned Medium In Vitro**. (**A**–**E**) 3T3-L1 cells were seeded on Day-3 with either βgal or hCTHRC1 conditioned medium at a 1/4 dilution. Whole-cell lysates were collected on Day 0 to determine SOX9 expression by qPCR (**A**) and Western blot (**B**) analyses. (**A**) Average fold expression differences in *Sox9* mRNA levels relative to housekeeping *Gtf2b* expression levels from five independent experiments (n = 5; ** *p* ≤ 0.01). Green symbols (triangle, circle, diamond, open square, and closed sqaure) are paired according to experimental replication. (**C**) Average SOX9 protein fold change densitometry values relative to housekeeping GTF2B protein expression levels from three independent experiments (n = 3; ** *p* ≤ 0.01). Green symbols (triangle, open circle, and closed circle) are paired according to experimental replication. (**D**,**E**) Representative confocal microscopy images of SOX9 protein localization on Day 0 in 3T3-L1 cells treated with either βgal conditioned medium (**D**) or hCTHRC1 conditioned medium (**E**): nuclei (blue); SOX9 (green); F-actin/Alexa Fluor 546 Phalloidin (red). Scale bar: 20 μm. (**F**–**H**) 3T3-L1 cells were lentivirally transduced with either an shRNA construct targeting *Sox9* mRNA (i.e., shSOX9), or a non-targeting scrambled control shRNA construct (i.e., shSCR). The resultant shSOX9 and shSCR cells were seeded on Day-3 with βgal or hCTHRC1 conditioned medium at a dilution of 1/60. Whole-cell lysates were collected from cohorts of shSOX9 and shSCR cells on Day 0 to assess SOX9 protein expression levels by Western blot analysis (**G**), while the other cohorts were chemically stimulated to undergo adipogenic differentiation for a total period of 6 days (**F**). (**F**) Representative Oil Red O quantification data. shSOX9 and shSCR cells were formalin fixed on Day 6 and stained with Oil Red O, which was then eluted and its concentration determined by absorbance spectroscopy. Per experiment, shSOX9 and shSCR cells were plated in 24-well plates in which 6 wells each were treated with βgal or hCTHRC1 conditioned medium at a dilution of 1/60 (n = 3; ** *p* ≤ 0.01). Not statistically significant (ns). (**H**) Average SOX9 protein fold change densitometry values relative to housekeeping GTF2B protein expression levels from three independent experiments (n = 3; * *p* ≤ 0.05). Green symbols (triangle, open circle, and closed circle) are paired according to experimental replication. Not statistically significant (ns).

**Figure 7 ijms-26-01804-f007:**
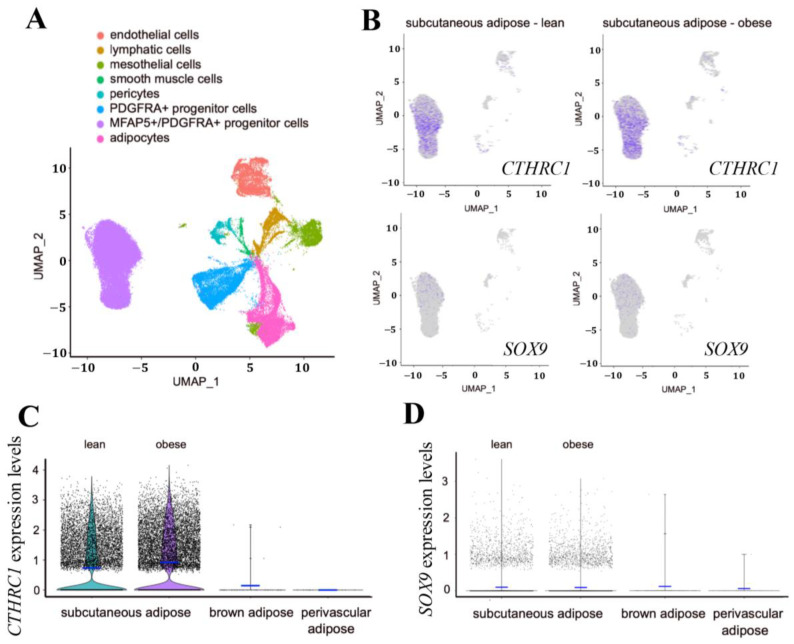
**Identification of *CTHRC1* and *SOX9* Gene Expression in *PDGFRA*^+^:*MFAP5*^+^ Enriched Cell Populations in Human Subcutaneous White Adipose Tissue**. (**A**) Harmonized UMAP projection of perivascular adipose and brown adipose single-nuclei RNA sequencing, as well as subcutaneous white adipose single-cell RNA sequencing, from lean and obese human donors. Data were filtered based on number of unique features and percent.mt. Doublets were removed using Scrublet. (**B**) UMAP projection displaying *CTHRC1* and *SOX9* expression as indicated by purple coloration. *CTHRC1* expression is observed in *PDGFRA*^+^:*MFAP5*^+^ cell populations in subcutaneous white adipose from both lean and obese human donors. *SOX9* mirrors *CTHRC1* gene expression patterns, though is expressed at lower levels. (**C**,**D**) ViolinPlots displaying the distribution of *CTHRC1* (**C**) and *SOX9* (**D**) expression among *PDGFRA*^+^:*MFAP5*^+^ cells in human perivascular, brown, and subcutaneous white adipose tissues. Mean expression is displayed by a blue horizontal bar. In comparing subcutaneous white adipose from lean versus obese human donors, there was no significant difference in the expression levels of *CTHRC1* or *SOX9*.

**Figure 8 ijms-26-01804-f008:**
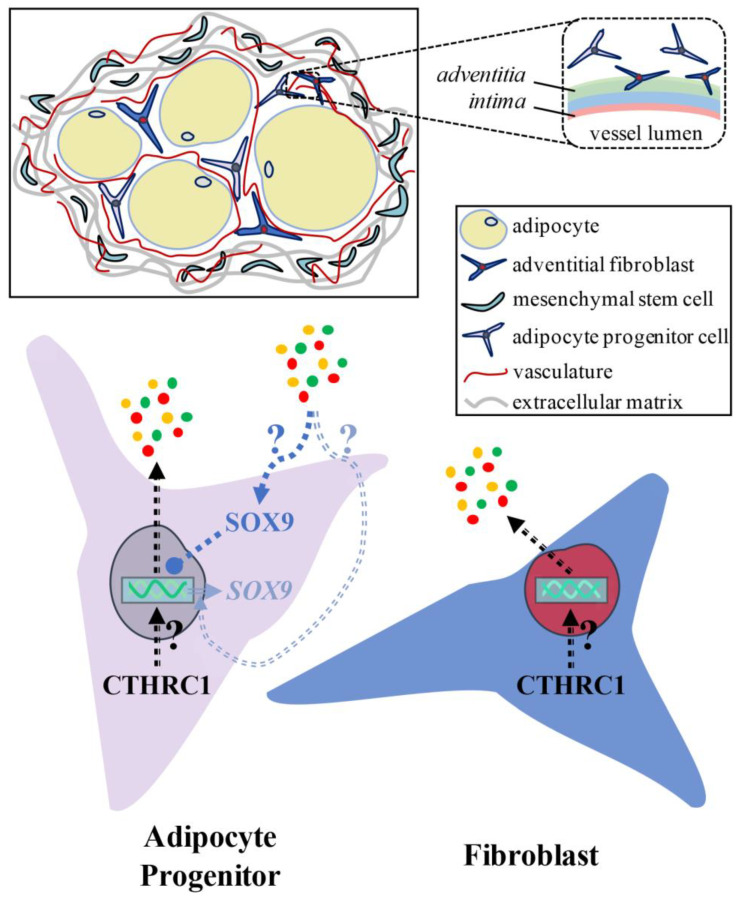
**Conceptualizing the CTHRC1-SOX9 Signaling Axis Within White Adipose Tissue**. Graphical illustration depicting white adipose tissue architecture and addressing the question of whether CTHRC1 is expressed in adipocyte progenitor cells and/or fibroblasts. Within white adipose tissue, adipocytes reside in a highly vascularized niche which contains adipocyte progenitor cells and fibroblasts that are each positioned between adipocytes and adjacent to vascular adventitia [22]. Merrick and colleagues previously identified the reticular interstitium which is an anatomically distinct, fluid-filled compartment that expresses a robust collagen-rich network of extracellular matrix and houses mesenchymal stem cells [23]. In brief, our study supports that CTHRC1 is expressed in adipocyte progenitor cells and/or fibroblasts within white adipose tissue. Intriguingly, although CTHRC1 itself is a secreted protein, our in vitro data could indicate that there is a secreted CTHRC1 effector that enhances both SOX9 gene expression and SOX9 protein nuclear translocation. To this end, it has been reported that SOX9 silences adipogenic transcription factor genes by binding to their promoters resulting in the inhibition of adipogenesis [12]. Collectively, our data suggest that CTHRC1 could regulate the secretome to enhance anti-adipogenic SOX9 signaling thus suppressing adipogenesis.

## Data Availability

Human PVAT single-nucleus RNA sequencing filtered feature barcode matrices were retrieved from the Gene Expression Omnibus data repository under accession GSE164528 [37]. Human deep-neck BAT single-nucleus RNA sequencing filtered feature barcode matrices were retrieved from the European Bioinformatics Institute under accession E-MTAB-8564 [37]. Filtered feature barcode matrices from single-cell RNA sequencing of human subcutaneous adipose tissue were retrieved from the Gene Expression Omnibus under accession GSE128890 [23]. Additional raw numerical data including uncropped immunoblot images used to prepare figures can be accessed via Figshare: https://doi.org/10.6084/m9.figshare.28038500.v2 (accessed on 15 January 2025). Data reporting is in accordance with the ARRIVE guidelines.

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
