# Peer review of "CTHRC1 Expression Results in Secretion-Mediated, SOX9-Dependent Suppression of Adipogenesis: Implications for the Regulatory Role of Newly Identified CTHRC1+/PDGFR-Alpha+ Stromal Cells of Adipose"

_ijms, 2025, doi:10.3390/ijms26051804_

Round 1
Reviewer 1 Report
Comments and Suggestions for Authors
The manuscript by Siviski et al examines role of CTHRC1 expression in SOX9 modulation of adipogenesis. These data also support a potentially novel cell population in WAT. The data clearly show that media from cells expressing human CTHRC1 inhibit adipogenesis of 3T3-L1 cells. The studies also show rigorous effects of exogenous human CHTRC1. The western blots are of very highly quality with appropriate controls. The need for SOX9 in this process was shown. A role of Rac and Rho was suggested by the use on inhibitors. Both CTHRC1 and SOX9 are shown to be in a subpopulation of progenitor cells from iWAT. Overall, this is a compelling study with rigorous data and the results are not over interpreted.
Additional comments for consideration:
There should be a title for supplemental Figure 1.
For Supplemental Figure 2, the title should be in bold.
Author Response
- There should be a title for supplemental Figure 1.
We thank the reviewer for their observation, and have included a title (in red text): Supplementary Supporting Data
- For Supplemental Figure 2, the title should be in bold.
Thank you and we fully agree. When we submitted both Microsoft Word and PDF versions of our manuscript to the journal, the manuscript was automatically reformatted to the version that you received for review. In some cases, this reformatting resulted in unintended changes to the font style in certain areas of the text as well as diminished resolution quality of the figures. These matters will be fixed and addressed with the journal before publication.

Reviewer 2 Report
Comments and Suggestions for Authors
The manuscript focuses on the CTHRC1 gene and its role as a secreted protein in adipogenesis, encompassing studies at the cellular level, animal models, and human tissues. The experimental design is reasonable and builds on a solid foundation. The molecular mechanisms of CTHRC1 in regulating adipogenic signaling are significant for studying human obesity and obesity-related diseases.
1. The title should be shorter and highlight the key points.
2. The full name of the SOX9 gene should be added in the Keywords.
3. Excessive descriptions of experimental methods should be avoided in the results section; for example, lines 105-107 should be removed.
4. Some data mentioned but not presented in the manuscript are suggested to be included in the supplementary files.
5. A scale should be added to the cell photographs in Figure 1F.
6. All figures are advised to be enhanced in resolution according to the submission requirements.
7. The entire text, including both the text and figures, should be reviewed to ensure uniformity in the formatting of gene names for the same species.
8. Remove one period at line 331.
9. It is suggested that the order of the images in Figure 4 be rearranged.
10. The effect that persists after immunodepletion is intriguing; it is recommended that the authors make bold speculations in the discussion.
11. Remove "tissue" at Line 789.
12. The lack of significant difference in CTHRC1 gene expression between obese and lean individuals' iWAT does not fully align with the researchers' expectations. This should be addressed in the discussion, specifying the possible reasons.
13. The selection of GTF2B as the internal reference gene by the authors has been observed. What was the rationale for opting for GTF2B instead of traditional internal controls? The authors need to provide an evaluation result of its validity as a reliable internal control.
Author Response
- The title should be shorter and highlight the key points.
We thank the reviewer for their suggested edit of the title. The title reflects the core findings of our study: i) CTHRC1 gene expression suppresses in vitro adipogenesis by positively regulating a secreted factor(s) that enhances the expression of SOX9; ii) identification of a novel CTHRC1+/PDGFR-alpha+ stromal cell population retained within subcutaneous white adipose tissue. The title is descriptive and, moreover, links together the in vitro and in vivo elements of our report.
- The full name of the SOX9 gene should be added in the Keywords.
Thank you, we have included the full gene name in the Keywords (in red text).
- Excessive descriptions of experimental methods should be avoided in the results section; for example, lines 105-107 should be removed.
We thank the reviewer for this recommendation and lines 105-107 have been removed.
- Some data mentioned but not presented in the manuscript are suggested to be included in the supplementary files.
Thank you and we wish to clarify that when we submitted both Microsoft Word and PDF versions of our manuscript to the journal, the manuscript was automatically reformatted to the version that you received for review. In result, all supplementary material was excised from the primary text and presented in a separate file. We would also like to point out that, in some cases, this automatic reformatting resulted in unintended changes to the font style in certain areas of the text as well as diminished resolution quality of the figures. These matters will be fixed and addressed with the journal before publication.
- A scale should be added to the cell photographs in Figure 1F.
Thank you for this suggestion. A scale has been added to this subfigure.
- All figures are advised to be enhanced in resolution according to the submission requirements.
Thank you, we fully agree. Please refer above to our response to critique no. 4.
- The entire text, including both the text and figures, should be reviewed to ensure uniformity in the formatting of gene names for the same species.
Thank you. Careful review has been conducted to ensure uniformity in gene nomenclature, particularly when referring to human or mouse species.
- Remove one period at line 331.
We thank the reviewer. This error was the result of the automatic reformatting of our text by the journal when we submitted the Microsoft Word and PDF versions of our manuscript. Prior to publication, we will carefully inspect the text to ensure there are no typos, font issues, or formatting errors of any sort.
- It is suggested that the order of the images in Figure 4 be rearranged.
We thank the reviewer for their recommendation but respectfully disagree. Figure 4 and the corresponding text introduce the rationale for investigating SOX9. First, we demonstrate that hCTHRC1 conditioned medium enhances SOX9 expression at both the mRNA and protein levels (A, B, and C). Next, we demonstrate that hCTHRC1 conditioned medium positively regulates SOX9 protein nuclear localization (D and E). Relevant to SOX9 protein nuclear localization, in the text we also refer to literature which demonstrated that the anti-adipogenic effect of SOX9 is based on its ability to bind the proximal promoter regions of critical adipogenic genes thus blunting adipogenic gene transcription. Finally, we justify our Sox9 gene knockdown (i.e., RNAi) studies on the basis that hCTHRC1 conditioned medium positively regulates both SOX9 gene expression and nuclear translocation. Accordingly, these RNAi-based experimental data are presented in the last three subfigures (F, G, and H) of Figure 4.
- The effect that persists after immunodepletion is intriguing; it is recommended that the authors make bold speculations in the discussion.
We thank the reviewer for this suggestion. These enhanced speculations are presented in the third paragraph of the Discussion (in red text).
- Remove "tissue" at Line 789.
We thank the reviewer and this redundancy has been removed.
- The lack of significant difference in CTHRC1 gene expression between obese and lean individuals' iWAT does not fully align with the researchers' expectations. This should be addressed in the discussion, specifying the possible reasons.
We thank the reviewer for this insight and refer them to paragraph five of the amended Discussion (see red text).
- The selection of GTF2B as the internal reference gene by the authors has been observed. What was the rationale for opting for GTF2B instead of traditional internal controls? The authors need to provide an evaluation result of its validity as a reliable internal control.
We thank the reviewer for this insightful question. GTF2B, a general transcription factor, is frequently used as a control housekeeping gene. One example includes the following published report (the cited article below includes a hyperlink for your reference):
SPT6 interacts with NSD2 and facilitates interferon-induced transcription. Ryota Ouda, Naoyuki Sarai, Vishal Nehru, Mira C. Patel, Maxime Debrosse, Mahesh Bachu, Răzvan V. Chereji, Peter R. Eriksson, David J. Clark, Keiko Ozato. FEBS Letters, 2018, Volume 592, Issue 10, May 2018 Pages 1681-1692.
In this report, Ouda et al. note, “Recruitment of SPT6 to ISGs depends on the interaction with NSD2. (A) ChIP-qPCR was performed for Ifit1, Stat1 (ISGs), and Gtf2b (housekeeping gene) for accumulation of SPT6 in WT (left) or NSD2 −/− (right) spontaneously immortalized cells.” Furthermore, Dr. Robert Koza, a co-author of our manuscript in review, has used GTF2B as a control in numerous published studies. Given the morphological changes that accompany preadipocyte-to-adipocyte differentiation, we avoided the use of cytoskeletal genes (e.g., actin, tubulin, etc.) for housekeeping purposes given that their expression levels can change during the course of adipogenesis.

Round 2
Reviewer 2 Report
Comments and Suggestions for Authors
The authors have addressed all the raised concerns and made the necessary revisions to the manuscript;